# Dietary Freeze-Dried Flaxseed and Alfalfa Sprouts as Additional Ingredients to Improve the Bioactive Compounds and Reduce the Cholesterol Content of Hen Eggs

**DOI:** 10.3390/antiox12010103

**Published:** 2022-12-31

**Authors:** Simona Mattioli, Alice Cartoni Mancinelli, Elisabetta Bravi, Elisa Angelucci, Beatrice Falcinelli, Paolo Benincasa, Cesare Castellini, Valeria Sileoni, Ombretta Marconi, Alessandro Dal Bosco

**Affiliations:** 1Department of Agricultural, Food and Environmental Sciences, University of Perugia, 06121 Perugia, Italy; 2Italian Brewing Research Centre, University of Perugia, 06126 Perugia, Italy; 3Department of Economics, Universitas Mercatorum, Piazza Mattei 10, 00186 Rome, Italy

**Keywords:** laying hen eggs, sprouts, phytoestrogens, vitamin D, sterols, antioxidants

## Abstract

Eggs are a complete food with high-quality proteins; a 2:1 ratio of unsaturated to saturated fatty acid (SFA); and a good amount of minerals, as well as vitamins or antioxidant compounds. Seeds or mature plants were usually added to the feed to improve egg quality. This study aimed to evaluate the effect of alfalfa and flax freeze-dried sprouts supplementation in diets of laying hens on egg oxidative status and key bioactive compounds. Thirty Sassò hens were fed with three different diets: standard, standard + 3% freeze-dried alfalfa sprouts, or flaxseed sprouts. Ten pools of 10 egg yolks per group were collected at 0, 4, and 8 weeks and analyzed. Supplementation with sprouts enriched the phytosterols, phytoestrogens, tocols, carotenes, vitamin D, and n-3 fatty acid contents in the eggs. Cholesterol content was lower in both sprout-supplemented groups, and a decrease in its oxidative products was also observed. It was found that a 3% freeze-dried sprouts supplementation of approximately 56 days improves the egg quality. Further studies are necessary to verify higher supplementing doses and the applicability of this strategy in the commercial egg production chain.

## 1. Introduction

Table eggs are largely consumed worldwide [1], with approximately 1140 billion eggs produced annually, and 700 billion/year, globally consumed [2]. They represent an inexpensive protein source (USD 0.30/g protein) [3]. From a nutritional point of view, table eggs are a complete food with high-quality proteins; a 2:1 ratio of unsaturated to saturated fatty acid (SFA); and a good amount of iron, phosphorus, and other minerals, as well as vitamins (i.e., vitamin A and E) or antioxidant compounds [1]. Eggs are also rich in phospholipids that can affect intestinal cholesterol absorption [2]; this is considered their limiting factor for human consumption (approximately 200 mg of cholesterol/g egg). Furthermore, eggs naturally contain vitamin D and are thought to contribute up to 10% of their dietary intake [4]. The United Kingdom Department of Health showed that one whole egg contains 3.15 µg of vitamin D/100 g, and additional research demonstrated that eggs could be potentially enriched in vitamin D [4]. Based on the positive characteristics reported herein and to obtain some “plus” qualitative features, many researchers have focused on strategies to further improve the quality of table eggs, for example, nutritional labels claim that they are “omega-3 fatty acids source” when eggs contain more than 15% of n-3 polyunsaturated fatty acids (PUFAs) or are “high in omega-3 fatty acids” when eggs contain more than 30% n-3 PUFA of the recommended nutritional intake for an adult male [5]. The most common enrichments of eggs are that of n-3 fatty acids [6,7], vitamin E [8], vitamin D [9], selenium [10], lutein [11,12], and phytoestrogens [13]. However, few studies have analyzed the interaction of such enrichments with other critical traits of eggs, such as lipid oxidation and cholesterol content. In a prior study [13], we demonstrated that dietary flaxseed supplementation (10%) in laying hens partially reduced egg cholesterol with respect to the control diet (189.14 vs. 195.22 mg/egg). Similarly, the use of dietary flaxseed or alfalfa sprouts slightly reduced the cholesterol content of eggs (11.5 vs.10.5 mg/g yolk, respectively, in control and sprouts-supplemented diets). Compared with seeds or mature plants, sprouts contain low amounts of antinutrients [14] and are rich in amino acids, fatty acids, and simple sugars as result of the enzymatic breakdown of large macromolecules and bioactive compounds, known as phytochemicals [6,15]. Despite the nutritional benefits of sprouts, their use is not always straightforward. Fresh sprouts have a short shelf life (3–4 days at 5 °C), which makes them highly perishable and susceptible to alteration and microbial proliferation (e.g., Escherichia coli, Salmonella enterica, and Vibrio cholerae) if not handled or stored properly [15]. Accordingly, nutritional changes occur easily, reducing nutraceutical value and quality. Thus, processing methods that extend the shelf life of sprouts should be considered. One effective method is the freeze-drying of sprouts, which slightly affects their oxidative status and thus better preserves their phytochemical compounds [16] as compared to fresh sprouts. Furthermore, with this strategy, a concentration of all bioactive compounds up to seven-fold is possible, suggesting a positive effect on egg quality when administered to laying hens. Based on these considerations, the objective of this study was to evaluate the effect of alfalfa and flax freeze-dried sprouts supplementation in the diets of laying hens on egg oxidative status (lipid and cholesterol oxidation) and key bioactive compounds (antioxidants, sterols, vitamin D, phytoestrogens, and PUFAs), with substantial attention toward the correlations among these compounds.

## 2. Materials and Methods

All chemicals and reagents used were of analytical grade and purchased from Sigma-Aldrich (Bornem, Belgium) unless specified otherwise.

### 2.1. Production of Alfalfa and Flax Sprouts

Alfalfa (*Medicago sativa* L.) and flax (*Linum usitatissimum* L.) seeds were prepared as reported in the previous paper [16]. For each species, sprouts obtained on the third day from different trays were pooled to prevent a possible tray effect and stored at 4 °C in plastic bags until freeze-dry procedure (i.e., within 3 days).

#### Drying Treatments

The sprouts were stored at −80 °C for 24 h and then lyophilized for 24 h under vacuum (6.67 Pa, Edwards freeze drying, Milan, Italy). Each sample was weighed before and after the thermal treatment to determine the water loss. After drying, 10 g of each sample was carefully minced and stored at −80 °C until analysis (1 week later), while the remaining material was used for the formulation of the experimental diets.

### 2.2. Animals and Diets

The experimental protocol was devised according to the Italian directives on animal welfare [17], following the International Guiding Principles for Research Involving Animals reported in the 2010/63/EU Directive and transposed into the 26/2014 Italian Legislative Decree. The research was conducted at the experimental farm of the Department of Agricultural, Food, and Environmental Science of the University of Perugia (Perugia, Italy).

Thirty commercial hens, Sassò type, aged 30 weeks at the start of the experiment, were randomly divided into three dietary groups of 10 hens as follows:Standard diet (C);Standard diet + 3% of freeze-dried alfalfa sprouts (A);Standard diet + 3% of freeze-dried flaxseed sprouts (F).

The hens were maintained in indoor pens under standard housing conditions, and an artificial photoperiod of 16 h of light and 8 h of dark was applied. The building was under a controlled ventilation regime (10 m^3^/hen/h), the temperature range was 23–27 °C, and the relative humidity was 50–80%.

Standard feed and water were provided ad libitum by manual bell feeders and automatic drinkers, respectively. (Table 1) to all groups, and the daily residues were weighed to evaluate the voluntary feed intake.

The experimental period lasted 56 days. Egg deposition was recorded daily; the eggs were collected ten days before the start of the experiment (baseline, day 0), in the middle (4th weeks.), and at the end (8th weeks.) of the experimental trials (Figure 1).

### 2.3. Eggs Sampling

For each dietary treatment, 63 eggs/week (total egg/group = 530 ± 26) to make 10 pools of 10 egg yolks during weeks 0 (baseline), 4, and 8 and stored at 5 °C until analytical processing (maximum 2 days later) were collected. All the following reported analytical determinations were performed on the 10 pools created.

### 2.4. Analytical Determination

The chemical composition of the feed and freeze-dried sprouts was determined according to the official methods [18]. Feed samples were analyzed in triplicate. The lipid proportion of egg yolk was also determined [18].

#### 2.4.1. Fatty Acids

Fatty acid profiles of the feed and egg yolk were determined using gas chromatography (GC) after lipid extraction, according to the method reported by Folch et al. [19] considering the water contents of the samples. To obtain fatty acid methyl esters, we dried the lipid extract with a rotavapor and added 1 mL n-hexane. Finally, the trans-methylation procedure was obtained with 0.5 mL 2 M KOH methanol solution at 60 °C for 15 min. The fatty acids were determined using a GC Varian (CP-3800, Agilent, Milan, Italy) equipped with an FID detector and a capillary column of 100 m length × 0.25 mm × 0.2 μm film (Supelco, Bellefonte, PA, USA). Helium was used as the carrier gas at a flow rate of 0.6 mL/min. The split ratio was 1:20. To calculate the amount of each fatty acid, we used heneicosanoic acid (C21:0, Sigma-Aldrich analytical standard, Milano, Italy) as the internal standard. The average amount of each fatty acid was used to calculate the sum of the total SFA, monounsaturated (MUFA), and PUFA. The relative proportion of individual fatty acids is expressed as a percentage for diets and as mg/100 g for eggs. The index of nutritional quality (INQ) was calculated as the amount (mg/100 g) of eicosapentaenoic acid (C20:5n-3, EPA), docosapentaenoic acid (C22:5n-3, DPA), and docosahexaenoic acid (C22:6n-3, DHA) with respect to the total energy (kcal/100 g) of the egg [20].

#### 2.4.2. Antioxidants

The different isoforms of vitamin E (α-, β + γ, and δ tocopherol-T, and tocotrienols-T3) in the diets and egg yolk were measured by HPLC [6]. Briefly, the lyophilized egg yolk or ground feed (0.1 g) was saponified in 1 M KOH in ethanol in a thermostat bath at 50 °C for 1 h. The contents were then extracted twice with n-hexane (10 mL). The upper phase was collected and dried with N_2_ to be reconstituted in 200 µL acetonitrile. A total of 50 μL was injected into the HPLC system (Perkin Elmer series 200, quaternary pump, equipped with an autosampler system, model AS 950-10, Tokyo, Japan) on a Sinergy Hydro-RP column (4 µm, 4.6 × 100 mm; Phenomenex, Bologna, Italy). The run length was 6 min, with an isocratic flow of 2 mL/min of a solvent mixture composed of acetonitrile/methanol/tetrahydrofuran/ammonium acetate 1% (65/21/11/3, *v*/*v*/*v*/*v*). The isoforms were identified using a fluorescent detector (model Jasco, FP-1525; excitation and emission wavelengths of 295 nm and 328 nm, respectively) and quantified using external calibration curves prepared with increasing amounts of pure tocols in ethanol.

The main carotenoids in the diets and yolk were determined using the same procedure and the HPLC system previously described. The solvent system consisted of solutions A (methanol/water/acetonitrile, 10/20/70, *v*/*v*/*v*) and B (methanol/ethyl acetate, 70/30, *v*/*v*). The volume of injection was 20 μL while the flow rate was 1 mL/min. An elution program was applied: from 90% A in a 20 min step to 100% B; then, a second isocratic step of 10 min, for a total run length of 30 min. The detector was a UV–visible spectrophotometer (Jasco UV2075 Plus) set at a wavelength of 450 nm for lutein, zeaxanthin, and β-carotene and 325 nm for retinol. Carotenoids were identified and quantified by comparing the samples with pure commercial standards diluted in chloroform (Sigma-Aldrich, Steinheim, Germany; Extrasynthese, Genay, France). Data were expressed as mg/100 g of eggs. The limit of detection (LOD), determined as three times the signal-to-noise ratio, and the limit of quantitation (LOQ), determined as ten times the signal-to-noise ratio, of each compound were also calculated. The LOD values (µg/mL) were retinol 0.003, lutein 0.022, zeaxanthin 0.010, β-carotene 0.009, δ-Tocopherol 0.012, γ-Tocopherol 0.020, and α-Tocopherol 0.015. The LOQ values (µg/mL) were retinol 0.011, lutein 0.062, zeaxanthin 0.033, β-carotene 0.020, δ-Tocopherol 0.045, γ-Tocopherol 0.060, and α-Tocopherol 0.055.

#### 2.4.3. Phytoestrogens

The extraction of lignans and isoflavones was performed according to Mattioli et al. [13]. Briefly, 0.5 g of freeze-dried egg yolk was extracted by refluxing in aqueous ethanol for 30 min and filtered through a Whatman filter paper. The ethanolic extract was made up to 50 mL with absolute ethanol and 5 mL was taken to dryness. The dried extract was reconstituted in acetate buffer (pH 4.5) and then subjected to hydrolysis at 37 °C overnight with a filtered solution of 10,000 Fishman Units of a mixed β-glucuronidase/sulphatase (*H. pomatia*, Sigma Chemicals Inc., Milano, Italy). After hydrolysis, lignans and isoflavones were isolated by solid-phase extraction on a C18 ODS SPE cartridge previously conditioned with methanol followed by distilled water. A total of 2 mL of incubated sample was then passed through the cartridge, which was washed with 5 mL of distilled water. Finally, the eluate was collected with 5 mL of methanol, taken to dryness, suspended in 2 mL of methanol, and injected. Isoflavones (daidzein, genistein, glycitein, equol), coumestans, and lignans (secoisolariciresinol diglucoside-SDG, secoisolariciresinol-SECO, matairesinol-MATA, pinoresinol-PINO, enterodiol-END, enterolactone-ENL, and coumestrol) were determined and quantified using UHPLC system equipment consisting in a Knauer 3950 autosampler with a 20 µL loop, a quaternary Azura P 6.1 L pump (Knauer, Berlin, Germany) coupled with an Azura MWD 2.1 L height channel UV–visible detector. The separation was carried out using a Luna Omega PS C18 column (Phenomenex Inc., Torrance, CA 90501-1430 USA, 50 mm 2.1 mm ID) at 40 °C and a flow rate of 0.4 mL/min. Mobile phase A was 50 mM sodium acetate buffer (pH 5) and methanol (80/20; *v*/*v*), and mobile phase B was phase A, methanol, and acetonitrile (40/40/20, *v*/*v*/*v*). The chromatographic separation was achieved using the following elution gradient: mobile phase A 70% (0 min), 70% (4 min), 50% (6 min), 70% (10 min), and 70% (12 min). The wavelengths used for the detection were 236, 254, 278, and 283 nm, and Clarity Chromatography Software for Windows (DataApex, Prague, Czech Republic) was used for data acquisition and elaboration. The external standard curve was used for calibration (a stock solution of 100 µg/mL of a mix of standard compounds in eluent A was used to prepare working solutions), and calibration plots were constructed for standard compounds with linearity between 0.15 and 1.5 µg/mL. The calibration curve for each standard compound was plotted using UV detection close to the maximum UV absorption for a given substance (236 nm for PINO; 254 nm for daidzein, genistein, glycitein, and coumestrol; 278 nm for END, MATA, and ENL; 283 nm for SECO, SDG, and equol). Data were expressed as µg/100 g of eggs. The LOD, determined as three times the signal-to-noise ratio, and the LOQ, determined as 10 times the signal-to-noise ratio, of each phytoestrogen were also calculated. The LOD values (µg/mL) for each phytoestrogen were as follows: SDG, 0.060; SECO, 0.102; PINO, 0.020; daidzein, 0.025; END, 0.012; MATA, 0.016; equol, 0.017; ENL, 0.021; genistein, 0.027; glycitein, 0.025; and coumestrol, 0.014. The LOQ values (µg/mL) were as follows: SDG, 0.340; SECO, 0.340; PINO, 0.068; daidzein, 0.083; END, 0.039; MATA, 0.052; equol, 0.57; ENL, 0.069; genistein, 0.091; glycitein, 0.093; and coumestrol, 0.047 µg/mL.

#### 2.4.4. Sterols and Vitamin D

The sterols in the diets and yolks were extracted with n-hexane as reported by Mattioli et al. [16] with the same procedure as for Vitamin E. Starting from 0.1 g of freeze-dried and finely ground sprouts or yolk, they were amilled using a Cyclotec 1093 sample Tecator to a particle size of about 1 mm. The extract was injected into the same HPLC system (10 µL) as previously described.

The quantification of sterols was performed with an analytical column of C18 reverse phase type (particle size ODS-2.5 M, 4.6 mm internal diameter; CPS Analytical, Milan, Italy) as specified in Mattioli et al.’s work [16]. The run length was 20 min and the solvent mixture was isocratic of acetonitrile/isopropanol (70/30; *v*/*v*). Sterols were identified using a UV detector (Model 2075 Plus Jasco, Tokyo, Japan) at λ 210 nm and quantified using a calibration curve with increasing amounts of pure standard solution (cholesterol, β-sitosterol, stigmasterol, campesterol, avenasterol, brassicasterol) in isopropanol. Data were expressed as mg/100 g of eggs. The LOD values (µg/mL) were cholesterol 0.010, β-sitosterol 0.020, stigmasterol 0.035, campesterol 0.012, avenasterol 0.025, and brassicasterol 0.030. The LOQ values (µg/mL) were cholesterol 0.030 β-sitosterol 0.062, stigmasterol 0.105, campesterol 0.038, avenasterol 0.085, and brassicasterol 0.090.

To quantify the main forms of vitamin D (cholecalciferol- vitamin D_3_ and 25-hydroxy cholecalciferol-25-OH vitamin D_3_), we used the same sterol extract and HPLC system previously described. The mobile phase consisted of acetonitrile/methanol (75/25, *v*/*v*) at a flow rate of 2 mL/min. The injected volume was 20 μL. The vitamin D forms were identified using a UV detector (Model 2075 Plus Jasco, Tokyo, Japan) at λ 280 nm and quantified using a calibration curve with increasing amounts of pure standards solution (Sigma-Aldrich, Steinheim, Germany) in ethanol. Data were expressed as μg/100 g of eggs. The LOD values (µg/mL) for vitamin D_3_ 0.025 and 25-OH vitamin D_3_ were 0.032. The LOQ values (µg/mL) were 0.076 for vitamin D_3_ and 0.093 for 25-OH vitamin D_3_.

#### 2.4.5. Oxidation Assays: Cholesterol-Oxidized Products and Thiobarbituric Acid Reactive Substance

The quantification of cholesterol-oxidized products (COPs) was performed using an HPLC/UV–visible system as reported for sterols. The same sterol extracts were used for COPs quantitation by modifying the HPLC program as follows. The mobile phase was composed of a mixture of acetonitrile/isopropanol (70:30, *v*/*v*) and released at a flow rate of 1.5 mL/min. The injected volume was 10 µL. The COPs were identified using a UV detector (Model 2075 Plus) set at 206 nm for 7-hydroxycholesterol (7-OH) and 233 nm for 5-cholesten-3b-ol-7-one (7-Keto) using a programmed step run of 20 min and quantified using a calibration curve with increasing amounts of pure standard solutions in isopropanol. Data were reported as µg/100 g of eggs. The LOD values (µg/mL) for 7-OH were 0.036 and 7-Keto 0.016. The LOQ values (µg/mL) were 7-OH 0.103 and 7-Keto 0.048.

Thiobarbituric acid reactive substance (TBARS) concentrations were assessed according to the method of Cherian et al. [21]. Briefly, 2 g of egg yolk samples were used and extracted with 18 mL of 3.86 g/100 mL perchloric acid. A total of 50 µL of butylated hydroxytoluene (4.5 g/100 mL ethanol) was added to each sample during extraction to control lipid oxidation, and the samples were centrifuged at 6000× *g* for 10 min. The homogenate was filtered through a Whatman No. 1 filter paper. Two milliliters of the filtrate were mixed with 2 mL of 0.3 g/100 mL thiobarbituric acid in distilled water and incubated in the dark at room temperature for 15–17 h. Absorbance was set at 531 nm wavelength. The results are expressed as µg MDA/100 g of eggs.

### 2.5. Statistical Analysis

Data were analyzed using a linear model (SPSS, v27) with diet, time of sampling, and their interaction as fixed effects. The significance of differences among least squares means was determined using the multiple t-test (*p* < 0.05). A matrix of Pearson product–time correlations was built to detect associations among the main variables (Σ antioxidants, cholesterol, Σ phytosterols, Σ COPs, Σ phytoestrogens, Σ vitamin D, and Σ n-3). The correlation was defined as high when the Pearson coefficient was (r) > |0.5|, medium when r ranged from 0.3 to 0.5, and low when r <|0.3|.

## 3. Results

Table 1 shows the formulation and chemical composition of the diets. The main fatty acids and bioactive compounds are shown in Table 2.

Sprout-enriched diets showed a similar amount of antioxidant compounds and fatty acid profile, except for α-linolenic acid (C18.3n-3, ALA), which was higher in the A and F diets than in the control.

Phytoestrogens showed a higher concentration, mainly in the F diet than in the control, due to the lignan richness of the flaxseed sprouts (mainly SDG). By contrast, isoflavones exhibited lower values in sprout-supplemented diets than in the control. Phytosterols exhibited higher concentrations in the A and F diets than in C, mainly due to β-sitosterol and stigmasterol, which were not detectable in the control group. In contrast with the other groups, the F diet contained brassicasterol.

Table 3 reports the antioxidant content, sterol profiles, vitamin D, and phytoestrogens content of egg yolk. Retinol was the main antioxidant and exhibited an increase due to the diet and the length of administration (time). The same trend was found for carotenes, where the most representative was zeaxanthin in groups A and F. Tocols showed an increasing concentration based on the diet administered until 4 weeks of feeding, whereas no further significant changes were observed at 8 weeks.

The cholesterol content of eggs decreased with time in groups A and F, whereas phytosterols increased in both supplemented groups. Campesterol exhibited higher values in group F than that in group A, and β-sitosterol was higher in group A after 8 weeks of feeding.

Vitamin D exhibited an increase after 4 weeks of administration, and the most representative form was cholecalciferol (vitamin D_3_).

Phytoestrogens (Table 3) were dependent on the sprouts administered. Among the lignans, daidzein presented values below the detectable limits in all groups, whereas genistein was recorded in the F group after 4 weeks of feeding and in both sprout groups after 8 weeks. Even isoflavones exhibited a low level; in particular, PINO was not detectable in all eggs, and MATA was not detectable in the C groups. A higher value of such isoflavones, mainly after 8 weeks of feeding, was found in the F than in the A eggs, similar to what was observed for coumestrol.

The oxidative status of the eggs was not affected by the dietary plan, but lipid oxidation slightly increased in groups A and F (Table 4) over time. Conversely, COPs exhibited lower values in the sprout-supplemented groups than in C and baseline level, mainly due to the 7-OH forms (two- and three-fold lower, respectively, at 4 and 8 weeks for both the sprout-supplemented groups).

Lipid content and fatty acid profiles are reported in Table 4. Flaxseed- and alfalfa-enriched diets did not affect the lipid percentage of egg yolk; moreover, they provided a similar amount of linoleic acid (C18:2n-6, LA). However, its content in the egg yolk was higher in F and A eggs with respect to the C eggs. Furthermore, the long-chain derivatives of n-3 (EPA, DPA, and DHA) increased in both treated groups, along with a reduction in ALA. Most notably, the F eggs showed higher values of DHA than the A and C eggs after 8 weeks of dietary supply (54.08 vs. 31.62 and 30.75; 55.47 vs. 32.43 and 31.25 mg/100 g of egg yolk). By contrast, LA and AA content largely decreased with sprout administration. No significant effects due to time or their interaction with the diet were recorded. The eggs of the F groups exhibited better IQN with respect to A and mainly to C.

Table 5 presents the correlation matrices for the main egg compounds. A positive correlation was found between antioxidants and phytosterols, phytoestrogens, and total vitamin D, whereas a negative correlation was found between cholesterol and total COPs. A positive correlation was also observed between phytosterols and phytoestrogens, vitamin D, and n-3 PUFAs.

Conversely, a negative correlation was found between cholesterol and phytosterols or phytoestrogens and their oxidative products (i.e., COPs) and phytoestrogen, vitamin D, and n-3. A negative correlation (*p* = 0.051) was observed between cholesterol and vitamin D_3_ levels.

## 4. Discussion

The use of freeze-dried sprouts in the diets of laying hens represents a good strategy to obtain eggs enriched with many bioactive compounds (n-3 PUFA, phytoestrogens, and vitamin D). Furthermore, they are easy to administer because they can be added as a powder in the standard feed. In a previous paper, we demonstrated that the freeze-dried procedure better preserves the nutritional characteristic of the sprouts, eliminating yeasts and bacteria growth, due to the removal of water, which in sprouts represented about 90% of proximate composition [16].

There is an enormous body of literature on how to enrich eggs with some compounds (e.g., flaxseed for increasing n-3 PUFA); however, no other compounds or the interaction between the several molecules involved was generally considered.

In this study, we aimed to analyze: (i) the effect of freeze-dried sprouts on egg nutritional quality and (ii) the interactive effect of these compounds.

The cholesterol content of eggs produced by hens fed freeze-dried sprouts was lower than that in the control after 28 days of experimental feeding, and the difference increased after 56 days, when the concentrations in A and F eggs were 131.21 and 138.16 mg/egg, respectively, against the 219.02 mg/egg of the C group. In our previous study [13], the administration of 40 g/d fresh alfalfa and flax sprouts slightly decreased cholesterol (approximately 9%), likely due to the modulation of cholesterol biosynthesis, suggesting a synergic effect of plant substances such as phytoestrogens, sterols, fatty acids, chlorophyll, polyphenols, and catechins. The Pearson matrix demonstrated that cholesterol was negatively correlated with antioxidants (*p* = 0.002), phytosterols (*p* = 0.001), phytoestrogens (*p* = 0.016), and vitamin D (*p* = 0.051), whereas the n-3 PUFAs correlation was not significant (*p* > 0.05).

Several plant compounds may have competitive effects on cholesterol metabolism. In particular, the great increase in phytosterols (campesterol and β-sitosterol) in F and A eggs, in contrast with what was found with fresh sprout administration [13], could have interfered with cholesterol adsorption, considering that the efficiency of intestinal absorption was cholesterol > cholestanol > campesterol > β-sitosterol [22]. The modulation of cholesterol absorption by phytosterols is probably complex and may involve the enterocyte and liver, as well as the intraluminal micelle, which displaces cholesterol from intraluminal intestinal micelles, reducing cholesterol uptake via the brush border membrane [23].

Similarly, isoflavones, lignans, and coumestans affect estrogen release. They are precursors of some hormone-like compounds [24] and provide health benefits to animals (increasing egg production) [25] and humans [26]. Morton et al. [27] measured the response of plasma lignans to dietary isoflavones or lignans (flaxseed) in hens and found that enterolignans were higher in the flax group, and the plasma concentration of ENL was 5 to 20 times higher than that of estradiol. In our prior study [6], the blood ENL/17β-estradiol ratio was 2:25 in control and flaxseed hens, suggesting a strong antagonism between these molecules.

In this study, the END concentration was detectable after 8 weeks of sprout supplementation, but not the ENL, which is likely due to the metabolization of the former. Setchell et al. [26] first proposed the production and metabolism of END and ENL by human fecal flora in vitro. The authors reported that SDG and MATA were metabolized to END by facultative bacteria and that ENL was produced from END through oxidation by other bacteria. Furthermore, the plant precursors of END and ENL that have been observed are SECO and its glycosides, MATA and its glycoside, lariciresinol, isolariciresinol, hydroxymatairesinol, arctigenin, pinoresinol, and syringaresinol [25].

In addition, PUFAs can contribute to the reduction in egg cholesterol [28], although the correlation was non-significant. Our results show that both sprouts decreased ALA, and increased EPA with respect to the control diet; an increase in DHA was found only in the eggs of the F group.

Strategies for increasing the n-3 PUFA content of eggs mainly consist of changing the rearing system (i.e., extensive system) [20,29] or adding ingredients rich in ALA (e.g., linseed and rapeseed) [6,30]. In addition, long-chain PUFAs (LC-PUFAs) may be added by including fish oils or algae [31,32]; however, the former strategy is more physiologically and environmentally sustainable because the fish sources, rich in LC-PUFAs, rapidly decrease due to decades of overfishing and the feed requirement of aquaculture [33].

This study confirms the efficiency of hens to elongate and desaturate ALA into n-3 LC-PUFAs [34,35] and to transfer it into the eggs, where they play a crucial role during chick embryo development [36]. Furthermore, it may also be hypothesized a preservative role of antioxidant molecules contained in the sprouts, toward egg LC-PUFAs, considering the higher perishability of such molecules [37].

However, the n-3 value obtained in this study was lower than what was required for nutritional claims. Health organizations and government agencies in most Western countries recommend the daily consumption of 600–1000 mg n-3 PUFA, 100–200 mg of which should be in the form of n-3 LC-PUFA (EPA + DHA). In this study, one egg per day provided 50–67 mg of n-3 fatty acids, which was much lower than the nutritional claim, indicating that a higher addition of freeze-dried sprouts (with respect to the 3% tested in this experiment) is likely necessary. Additionally, the sprout-supplemented eggs exhibited a great reduction in n-6 PUFA. This finding is important from a nutritional point of view, considering that many epidemiological and clinical studies [38] have established that n-6 fatty acids (mainly AA) are precursors of molecules (2-series prostaglandins, PGE2, and 4-series leukotriene, LTB4) that exert a higher pro-inflammatory effect than the 3-series ones (PGE3). Theoretically, lowering the concentration of AA in favor of EPA may trigger a decreased production from specific immune cell types (monocytes, neutrophils, and eosinophils) of AA-derived mediators of inflammation and a diminished ability of platelets to produce pro-thrombotic molecules. However, it is reported that increasing EPA and reducing AA resulted in the production of anti-inflammatory or even less inflammatory compounds and thromboxanes, with lower pro-aggregatory and vasoconstrictive properties [38].

Moreover, the increase in antioxidants in egg yolks, either as isoflavones [39] or as tocols and carotenes, is notable. Indeed, high levels of LC-PUFA, if not adequately protected, can transform LC-PUFA benefits into risks due to the formation of oxidative products. In our study, the amount of tocols and carotenes in eggs was two- (4 weeks) to three- (8 weeks) fold higher in sprout-enriched diets.

Tocols and carotenes are relevant antioxidant molecules that always work synergically. α-Tocopherol, the most abundant vitamin E isoform, has a chain-breaking action and can be regenerated from its oxidized form by ascorbic acid, carotenoids, and phospholipids [40]. Carotenes, including γ-carotene, lycopene, and lutein, exhibit high-power scavenging of free radicals and are among the main elements of defense against coronary heart disease [41]. In particular, lutein and zeaxanthin have a filter effect and protect the macula from degenerative, oxidative, and photochemical damage [42] and are also considered pro-vitamin A, while α-, β-, and γ-carotenes have vitamin A activity.

These compounds are highly pigmented (i.e., yellow, orange, and red), and present in fruits and vegetables; when consumed by birds, they are incorporated into the egg yolk. The dietary administration of these compounds improves the oxidative stability of the eggs [43]. Furthermore, a strong diet × time interaction was recorded. The carotenes increased in both supplemented groups, with a time-dependent trend, whereas the retinol decreased only in the A group. This was probably due to a compensation mechanism of the carotenes; indeed, considering that carotenes are precursors of retinol [44] and they are furnished mainly by the alfalfa, it is possible that the conversion rate was down-regulated.

Hens fed freeze-dried sprouts exhibited reduced lipid oxidation and COPs. Indeed, the amount of antioxidants was negatively correlated with COPs, whereas the correlation between cholesterol and COPs was positive, probably because the decrease in COPs could also be attributed to substrate (cholesterol) limitation [45].

Oxidative processes that occur on cholesterol can induce large alterations in yolk membrane composition and egg physical state, reducing the technological property of eggs. Furthermore, COPs trigger an oxidation cascade that can also damage other lipid molecules (i.e., LC-PUFA), leading to a more rapid deterioration of the egg [46].

The increase in total vitamin D in eggs of hens fed freeze-dried sprouts was also notable. From a research perspective, eggs have garnered significant attention as vehicles for the vitamin D intake of humans [4]. Eggs naturally contain vitamin D and are thought to contribute up to 10% of its dietary intake [9]. Raw eggs contain approximately 3 μg of vitamin D per 100 g of egg [9,47], with several variations due to the rearing system of hens (in free-range system, this reaches 5–6 μg/100 g egg) [47]. In this regard, although the recommended human intake of vitamin D varies for the different age groups, in Europe, the guidelines for adults are between 10 and 20 μg/day [5]; hence, an egg portion (2 medium eggs) may cover more than 1/3 of the nutritional requirements [9].

In this study, the vitamin D concentration was close to the abovementioned value; however, its concentration was only slightly higher than that of the control egg.

Data on vitamin D and cholesterol demonstrated that dietary sprout reduced cholesterol content and increased vitamin D, especially when flaxseed was administered. It is reported that cholesterol is a precursor of vitamin D; indeed, it can be obtained either through food or sun exposure, where sun exposure is the major contributor to vitamin D in humans. It is possible that the reduction in cholesterol in the egg yolks was due to some metabolic processes that occurred in the hens, converting it into vitamin D [48].

Currently, the vitamin content of eggs is furnished by adding “synthetic” additives to the feed of laying hens [49]. In this context, a process that consumers may perceive as more “natural” than traditional fortification [9] may be beneficial, and that could be provided through supplementation with freeze-dried sprouts.

## 5. Conclusions

The data herein reported demonstrate that the 3% administration of freeze-dried sprouts as feed ingredients for laying hens enriched the eggs in many bioactive compounds (i.e., phytosterols, phytoestrogens, tocols, carotenes, vitamin D, and n-3 PUFA) after 56 days of trial. Moreover, a decrease in oxidative products (i.e., COPs) and thus an improvement in the oxidative status (TBARS) were observed. A notable reduction in yolk cholesterol (almost 100 mg/100 g egg yolk) was also found, which represents a further nutritional goal. However, some positive modifications, although significant, are minimal.

Further studies are necessary to verify the possibility of increasing the supplementing dose, the administration time (or even both solutions jointly), and the applicability of this dietary strategy in the egg production chain.

## Figures and Tables

**Figure 1 antioxidants-12-00103-f001:**
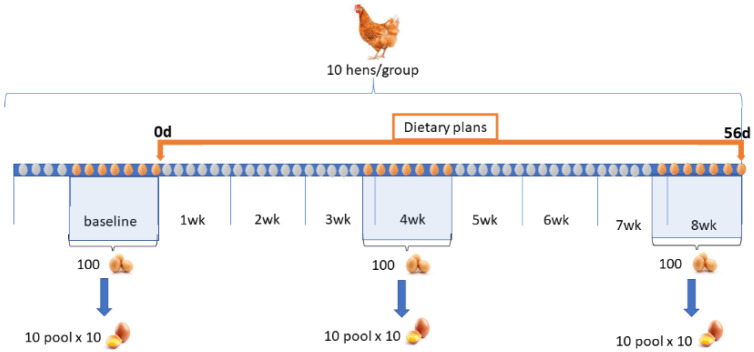
Dietary plan of the experimental trial. Nine eggs/group/day, for a total of 66 days (10 days before dietary administration + 56 days experimental period) were collected. One hundred eggs from 0 (baseline), 4, and 8 weeks were stored and pooled (10 pools of 10 eggs) for analytical determination.

**Table 1 antioxidants-12-00103-t001:** Mean ingredients (g/kg) and nutrient composition (g/kg) of control and alfalfa or flax freeze-dried sprouts-enriched diets.

Ingredients	C	A	F
Corn	450	450	450
Extruded soybean flakes	200	200	200
Maize gluten feed	160	160	160
Sunflower meal	88	88	85
Alfalfa meal	30	27	30
Freeze-dried alfalfa sprouts	-	3	-
Freeze-dried linseed sprouts	-	-	3
Vitamin mineral premix †	10	10	10
Calcium carbonate	50	50	50
Dicalcium phosphate	5	5	5
Sodium bicarbonate	5	5	5
Salt	2	2	2
Nutrient composition			
Water	110	122	118
Crude protein	178	179	178
Ether extract	53	54	56
Crude fiber	56	56	55
Ash	111	111	112

(C, Standard diet; A, Standard diet + 3% of freeze-dried alfalfa sprouts; F, Standard diet + 3% of freeze-dried flaxseed sprouts). † ZAGRO, provided per kg of diet: vitamin A, 12 500 IU; cholecalciferol, 3000 IU; DL-α-tocopheryl acetate, 60 mg; vitamin B_1_, 2 mg; vitamin B_2_, 6 mg; vitamin B_6_, 4 mg; pantothenic acid, 8 mg; PP, 30 mg; folic acid, 0.50 mg; vitamin B_12_, 0.02 mg; vitamin K, 2 mg; choline, 750 mg; Fe (sulphate monohydrate), 35 mg; Zn (zinc oxide), 42 mg; I (calcium iodate), 0.5 mg; Co (cobalt sulphate), 0.5 mg.

**Table 2 antioxidants-12-00103-t002:** Main fatty acids (% total fatty acids), antioxidant compounds (mg/kg f.m.), phytoestrogens (mg/kg f.m.), and phytosterols (mg/kg f.m.) in the control and 3% alfalfa (A) or flax freeze-dried (F) sprouts-enriched diets (mean + SEM).

	Experimental Diets		
	C	A	F
Fatty acids			
C16:0	17.25 ± 1.12	16.43 ± 0.88	16.20 ± 1.08
C18:0	21.12 ± 2.41	20.41 ± 1.09	20.45 ± 2.47
C18:1n-9	17.50 ± 1.02	17.48 ± 0.87	17.30 ± 1.04
C18:2n-6	22.25 ± 3.04	22.41 ± 2.74	21.78 ± 2.49
C18:3n-3	16.02 ± 0.98	18.10 ± 1.17	19.25 ± 1.97
Antioxidants			
α-tocopherol	120.45 ± 13.47	124.55 ± 12.39	123.85 ± 11.98
α-tocopheryl acetate	50.85 ± 6.12	52.30 ± 4.97	52.30 ± 5.01
γ-tocopherol	0.45 ± 0.04	1.55 ± 0.47	1.18 ± 0.68
δ-tocopherol	3.25 ± 0.12	3.11 ± 0.09	3.05 ± 2.98
α-tocotrienol	3.99 ± 0.09	4.85 ± 0.74	4.82 ± 0.45
γ-tocotrienol	5.65 ± 1.74	6.78 ± 1.47	6.75 ± 5.98
Σ Tocols	184.64 ± 22.64	193.14 ± 20.74	191.95 ± 19.47
Retinol	58.12 ± 6.12	57.44 ± 5.71	59.85 ± 6.47
Lutein	6.12 ± 1.04	9.34 ± 1.14	6.25 ± 0.98
Zeaxanthin	17.25 ± 1.70	39.86 ± 6.47	20.48 ± 1.98
β-carotene	11.78 ± 0.61	15.06 ± 1.14	10.85 ± 1.15
Phytoestrogens			
SDG	415.80 ± 23.51	415.15 ± 23.36	1761.21 ± 361.20
ISO	1.22 ± 0.13	2.90 ± 0.11	31.34 ± 2.74
SECO	n.d.	1.84 ± 0.23	36.84 ± 1.54
MATA	n.d.	0.41 ± 0.10	21.36 ± 3.22
Σ Lignans	417.12 ± 2.55	420.30 ± 28.71	1850.75 ± 194.70
Daidzein	70.61 ± 7.11	75.10 ± 3.14	74.71 ± 14.79
Genistein	39.82 ± 2.74	40.22 ± 4.79	40.22 ± 0.86
Glycitein	26.55 ± 0.61	25.12 ± 2.74	25.11 ± 0.91
Σ Isoflavones	153.67 ± 11.54	142.47 ± 13.61	140.04 ± 13.61
Coumestrol	n.d.	2.03 ± 0.11	42.67 ± 3.22
Phytosterols			
β-sitosterol	78.41 ± 7.11	82.71 ± 10.43	91.41 ± 10.54
Stigmasterol	n.d.	25.46 ± 8.55	13.21 ± 3.64
Campesterol	43.64 ± 1.42	42.54 ± 9.74	48.35 ± 8.71
Avenasterol	40.3 ± 3.22	47.11 ± 36.90	35.54 ± 6.33
Brassicasterol	n.d.	n.d.	9.37 ± 0.26
Σ Sterols	162.35 ± 20.90	197.82 ± 27.71	197.88 ± 37.44

Each value represents the mean of three replications. n.d., not detectable; f.m., fresh matter; SDG, secoisolariciresinol diglucoside; ISO, Isolaricilresinol; SECO, secoisolariciresinol; MATA, matairesinol. LOD: SECO, 0.102 µg/mL; MATA, 0.016 µg/mL; coumestrol, 0.014 µg/mL; stigmasterol, 0.035 µg/mL; brassicasterol, 0.090 µg/mL.

**Table 3 antioxidants-12-00103-t003:** Antioxidants (mg/100 g egg f.m.), sterols (mg/100 g egg f.m.), vitamin D (μg/100 g egg f.m.), and phytoestrogens (µg/100 g egg f.m.) content of eggs yolk, sampled at 0, 4, and 8 wks. of control (C) and 3% alfalfa (A) or flax freeze-dried (F) sprouts-enriched diets.

	Baseline	4 wk			8 wk			RMSE	*p* Value		
		C	A	F	C	A	F		Diet	Time	Diet × Time
Antioxidants											
Retinol	215.99	277.31	598.30	317.79	252.08	550.38	407.15	3.84	<0.001	0.002	0.001
Lutein	8.74	8.67	14.03	9.88	8.75	19.05	12.92	0.63	<0.001	<0.001	<0.001
Zeaxanthin	8.51	8.55	16.42	23.97	8.35	28.24	28.81	0.93	<0.001	<0.001	<0.001
β-carotene	0.95	1.00	1.73	1.07	0.96	1.64	1.10	0.18	0.002	0.021	0.037
Σ Carotenes	18.21	18.23	32.19	34.94	18.06	48.94	42.84	1.08	<0.001	<0.001	<0.001
δ-Tocopherol	0.58	0.34	1.04	1.22	0.77	1.17	1.06	0.13	0.013	0.349	0.375
γ-Tocopherol	1.82	2.33	3.32	2.77	2.61	2.30	2.42	0.20	0.235	0.305	0.454
α-Tocopherol	22.97	28.06	94.04	71.40	28.49	90.74	78.51	1.62	<0.001	0.177	0.720
Sterols											
Cholesterol, mg/egg	186.07	181.85	169.30	151.31	219.02	131.21	138.16	5.93	0.038	0.025	0.063
Cholesterol, mg/100 g egg	310.11	303.08	282.16	252.18	365.03	218.68	230.26	12.35	0.018	0.01	0.050
Campesterol	2.86	4.88	5.77	6.89	2.64	5.95	10.18	1.64	<0.001	<0.000	0.001
β-sitosterol	0.11	0.17	1.18	1.42	0.59	1.95	1.29	0.70	<0.001	<0.001	0.041
Vitamin D											
Vitamin D_3_	1.40	1.41	1.59	1.54	1.41	1.64	1.69	0.14	0.025	0.012	0.667
25-OH Vitamin D_3_	0.46	0.50	0.62	0.59	0.50	0.67	0.62	0.11	0.019	0.081	0.438
Σ Vitamin D	1.85	1.90	2.20	2.13	1.90	2.31	2.30	0.16	0.008	0.011	0.434
Lignans											
Daidzeina	n.d.	n.d.	n.d.	n.d.	n.d.	n.d.	n.d.	-	-	-	-
Genistein	n.d.	n.d.	n.d.	32.98	n.d.	41.82	58.32	5.09	<0.001	<0.001	<0.001
Isoflavones											
PINO	n.d.	n.d.	n.d.	n.d.	n.d.	n.d.	n.d.	-	-	-	-
MATA	n.d.	n.d.	76.92	116.76	n.d.	170.93	192.11	7.23	<0.001	<0.001	<0.001
Coumestrol	38.85	37.23	42.01	51.07	37.23	65.74	97.74	4.93	<0.001	<0.001	<0.001
Metabolites											
ENL	n.d.	n.d.	n.d.	n.d.	n.d.	n.d.	n.d.	-	-	-	-
END	n.d.	n.d.	n.d.	n.d.	n.d.	43.88	35.90	2.38	<0.001	<0.001	<0.001
Equol	87.31	89.69	119.39	164.49	89.69	224.32	262.30	8.46	<0.001	<0.001	<0.001

n = 10 pools of 10 egg yolks/per group; f.m.: fresh matter; RMSE: root mean square error. Vit D: cholecalciferol; 25-OH Vit D: Hydroxycholecalciferol. n.d.: not detectable. PINO, pinoresinol; MATA, mataresinol; ENL, enterolactone; END, enterodiol. Each value represents the mean of three replications. LOD: daidzein, 0.025 µg/mL; genistein, 0.027 µg/mL; PINO, 0.020 µg/mL; MATA, 0.016 µg/mL; ENL, 0.021 µg/mL; END, 0.012 µg/mL. d.m.: dry matter.

**Table 4 antioxidants-12-00103-t004:** Yolk lipid (%) content, cholesterol-oxidized products (COPs, µg/100 g egg), lipid oxidation (µg MDA/100 g egg), main polyunsaturated fatty acids (mg/100 g egg f.m.), and estimated INQ index of egg, sampled at 0, 4, and 8 wks, of control (C) and 3% alfalfa (A) or flax freeze-dried (F) sprouts-enriched diets.

	Baseline	4 wk			8 wk			RMSE	*p* Value		
		C	A	F	C	A	F		Diet	Time	Diet × Time
Lipids	30.3	31.24	30.56	30.12	30.97	30.27	30.22	1.81	0.525	0.765	0.562
Oxidation											
7-OH	123.87	131.05	59.38	53.24	128.09	42.34	45.06	16.05	<0.001	<0.001	0.002
7-Keto	6.25	4.92	4.37	5.47	7.38	3.22	4.74	1.24	0.559	0.977	0.126
Σ COPs	130.13	135.97	63.75	58.70	135.47	45.56	49.80	6.18	<0.001	<0.001	0.002
TBARS	114.37	93.18	112.50	96.09	105.79	110.76	124.23	11.58	0.057	0.014	0.005
Fatty acids											
18:2n-6, LA	801.49	827.09	208.83	217.57	877.87	203.73	212.27	12.20	<0.001	0.270	0.700
18:3n-3, ALA	44.80	46.93	24.96	31.10	40.96	25.60	31.89	1.30	<0.001	0.365	0.827
20:4n-6, AA	135.47	133.33	20.77	16.40	136.85	20.27	16.00	1.21	<0.001	0.276	0.785
20:5n-3, EPA	3.09	3.09	12.17	11.96	3.09	12.48	12.27	0.60	<0.001	0.144	0.702
22:5n-3, DPA	2.97	2.76	11.44	13.00	2.97	11.73	13.33	1.12	<0.001	0.349	0.915
22:6n-3, DHA	31.25	30.72	31.62	54.08	31.25	32.43	55.47	1.44	<0.001	0.332	0.646
INQ	0.29	0.29	0.43	0.62	0.29	0.44	0.63	0.14	<0.001	0.150	0.649

n = 10 pools of 10 egg yolks/per group; f.m., fresh matter; RMSE: root mean square error; 7-OH, 7-hydroxycholesterol; 7-KETO, 5-cholesten-3β-ol-7-one; COPs, cholesterol-oxidized products; TBARS, thiobarbituric acid reactive substances. INQ, index of nutritional quality; EPA, eicosapentaenoic acid; DPA, docosapentaenoic acid; DHA, docosahexaenoic acid.

**Table 5 antioxidants-12-00103-t005:** Matrix of Pearson correlation.

	Σ Antioxidants	Cholesterol	Σ Phytosterols	Σ COPs	Σ Phytoestrogens	Σ vitamin D	Σ n−3
Σ Antioxidants	1	**−0.547 ****	**0.628 ****	**−0.777 ****	**0.598 ****	**0.580 ****	0.059
		0.002	<0.001	<0.001	<0.001	0.001	0.758
Cholesterol		1	**−0.595 ****	**0.613 ****	**−0.437 ***	−0.265 ^+^	−0.294
			0.001	<0.001	0.016	0.051	0.115
Σ Phytosterols			1	**−0.759 ****	**0.843 ****	**0.691 ****	**0.510 ****
				<0.001	<0.001	<0.001	0.004
Σ COPs				1	**−0.797 ****	**−0.687 ****	**−0.404 ***
					<0.001	<0.001	0.027
Σ Phytoestrogens					1	**0.695 ****	**0.503 ****
						<0.001	0.005
Σ Vitamin D						1	0.306
							0.100
Σ n−3							1

** *p* < 0.01; * *p* < 0.05; ^+^
*p* = 0.05.

## Data Availability

Data is contained within the article.

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
