# Peer review of "Dietary Freeze-Dried Flaxseed and Alfalfa Sprouts as Additional Ingredients to Improve the Bioactive Compounds and Reduce the Cholesterol Content of Hen Eggs"

_antioxidants, 2022, doi:10.3390/antiox12010103_

Round 1
Reviewer 1 Report
very good data for the reader.
maybe other two treatments just the seeds added is better.
the birds used is limited in this trail.
Author Response
Dear Reviewer 1,
Thanks for the positive rating. We have decided not to provide the seed as well because many publications (even our previous ones) have analyzed the seeds that are a common ingredient; however a multiple comparisons, as suggested by the reviewer, could be interesting in the future.
As regards the number of hens considering that the experimental unit is the egg (for analytical determinations) and considering the EU directives on the use of animals for experimental purposes and the respect for animal welfare, it was decided to use the minimum number of hens needed to obtain a sufficient number of eggs for analytical evaluations. In the case of establishing the production performance with a more robust experimental design, we will examine a greater number of animals (see general comment).
Reviewer 2 Report
Dietary freeze-dried flaxseed and alfalfa sprouts as additional ingredients to improve the bioactive compounds and reduce the cholesterol content of hen eggs
This study evaluates the effect of freeze-dried sprouts on the yolk composition. The paper is well written and provides interesting information. Authors wrote a similar prior article including different information. In general, introduction gives enough information for readers but material and methods, results and discussion need to be improve. In material and methods section, authors should provide further information of some methodologies and on sample collection. Tables have also some mistakes and discussion must be improved in some aspects.
Abstract.
Line 14. as well as vitamins or antioxidants compounds (i.e., vitamin A and E).
(i.e., vitamin A and E) this are vitamins, should be after vitamins. Vitamin A is also consider antioxidant?
Line 20. A decrease in oxidative products was observed.
Could you please indicate in which groups?. Could you clarify this sentence?
Line 32. and other minerals, as 31 well as vitamins or antioxidants compounds (i.e., vitamin A and E)
Move (i.e., vitamin A and E) after vitamins as indicated before or delete it.
Materials and Methods
Lines 116-117. For each dietary treatment, 10 pools of 10 egg yolks were collected during weeks 0, 116 4, and 8 and stored at 5 °C until analytical processing (maximum 2 days later).
Please indicate how many eggs per treatment. How many eggs did you collected per day and how many days?. Where eggs collected different days distributed for different determinations? Or did you pool the eggs of different days and make same determinations in the pooled mixture. You should clarify the sample collection information.
Line 149. FD.
Did you mean fluorescent detector?.
Lines 142-151-Did you use a quaternary pump for HPLC of antioxidants?. Please indicate the total running time. Also indicate the solvents used and program.
Lines 152-167. Indicate run time for carotenoids. How do you make the sample extraction?. Did you use the same procedure based on saponification described for vitamins?
Linjes 171-190. Indicate how did you make the extraction of phytoestrogens. Did you use the same procedure based on saponification described for vitamins?
Lines 192-194. The sterols in the diets and yolks were extracted with n-hexane as reported by Mattioli et al. [16]. Starting from 0.1 g of freeze-dried and finely ground sprouts or yolk.
How did you ground the sample. Particle size?, or apparatous used (vortex?, manually? Etc)
Lines 192-204- Please describe briefly the extraction procedure or was the same as described for tocopherols?. Please indicate the run time for sterols, the solvents used for the mobile phase and the flow rate. Also, indicate volumen inyected.
Lines 213-214. Change D3 by D3 (subíndice). Check though out the text.
Lines 228-229. How did you make the extraction from samples. Did you homogenized them?.
RESULTS
Table 1. Maize. Do you mean corn?
For Feed, Alfalfa and Flax. Indicate abbreviation and define in table foot.
Indicate the form of Zn, Fe, I and choline added in the premix.
Check subíndices for B1, B2, B6, B12
Table 2.
a-tocopheryl acetate instead a-tocopherol acetate
Vitamin D in diets is missing
Lines 270-272. Indicate in which table are these data presented
…was higher in the A and F diets than in control. Did you test it statistically. P value is not presented in table
Line 290. Vitamin D3. Check subíndice
Table 3 instead Tabella 3.
The sum of carotenes it seems include Retinol. May be you can present retinol after the sum of carotenes
Check subindices for table 3. Colecalcipherol??. Should be cholecalcipherol?. The same for hydroxycolecalcipherol
Table 4 instead Tabella 4.
Change Yolk oxidative status of colesterol by “colesterol oxidative products” which is the definition of COPs
Change “significance” by P value
Define in the foot of the table RMSE and add superindices of definitions provided in foot table.
Line 335. D3 correct subíndice
Table 5 instead Tabella 5. Cholestol??. Do you mean cholesterol??
Figure 1 instead Figure 2. Why did you present this regression between cholesterol and vitamin D?. Cholesterol was negatively correlated with other parameters with lower P value. I would recommend to present a table with the parameters of the regression. Did you check if slopes were statistically different between treatments? You could make this comparison and include information in table that you built including R2 and P of the equation. Because for these parameters correlations were 0.051 but may be regression adjustments was not statistically significant.
DISCUSSION
It is missing if all the compounds tested are within values found in literature. Check with references.
Lines 369-371. The Pearson matrix demonstrated that cholesterol was negatively correlated with antioxidants (P = 0.002), phytosterols (P = 0.001), phytoestrogenes (P = 0.016), vitamin D (P = 370 0.05), and n-3 PUFAs (P > 0.05).
But correlation between colesterol and n-3 fatty acids was not statistically significant so there was not correlation between these variables.
Lines 397-400. In addition, PUFAs can contribute to the reduction of egg cholesterol [28], although the correlation was non-significant. Our results showed that both sprouts decreased ALA, and increased EPA respect to the Control diet; an increase in DHA was found only in eggs of F group.
Why dont you present long chain PUFA in the composition of the diets?. You give importance to these fatty acids and these were affected by dietary treatments but you only present C18:2 and C18:3.
Lines 407-409. This study confirms the efficiency of hens to elongate and desaturate ALA into n-3 407 LC-PUFAs [34,35] and to transfer it into the eggs, where they play a crucial role during 408 chick embryo development [36].
You should include the desaturase index to affirm this. Why are you sure these were not from the diets?.
Could be these higher long PUFA proportion found by sprouts due to higher antioxidant contents and protection. Clarify.
Lines 432-442. Tocols and carotenes are relevant antioxidant molecules often combined to exploit 432 their different properties. α-Tocopherol, the most abundant vitamin E isoform, has a 433 chain-breaking action and can be regenerated from its oxidized form by ascorbic acid, 434 carotenoids, and phospholipids……………..
Please, include which treatment do you think is better to improve antioxidants content.
How do you explain the time x diet interaction?. Retinol and tocols decrease in treatment A with time but increased in F. Could be this effect by interaction between different component??. Review in literature.
Lines 443-446. Hens fed freeze-dried sprouts exhibited reduced lipid oxidation and COPs. Indeed, 443 the amount of antioxidants was negatively correlated with COPs, whereas the correlation 444 between cholesterol and COPs was positive, probably because the COP decrease was also 445 due to the reduction of cholesterol and thus attributed to substrate limitation [42].
Include more information of the negative effects of COPS
Lines 447-449. The increase in total vitamin D in eggs of hens fed freeze-dried sprouts was also no- 447 table. From a research perspective, eggs have garnered significant attention as vehicles for vitamin D bio-fortification or enrichment [4]. Eggs naturally contain vitamin D and are 449 thought to contribute up to 10% of dietary intake [9]. Raw eggs contain approximately 3 450 μg of vitamin D per 100 g of egg [9,43], w……
Include recommended values of vitamin D for humans.
Lines 455-456. Linear regression of vitamin D and cholesterol demonstrated that sprout treatment 455 reduced cholesterol content and increased vitamin D, especially when flaxseed was ad- 456 ministered
How do you explain this effect?
Author Response
antioxidants-2057688
Reviewer 2
Comments and Suggestions for Authors
Dietary freeze-dried flaxseed and alfalfa sprouts as additional ingredients to improve the bioactive compounds and reduce the cholesterol content of hen eggs
This study evaluates the effect of freeze-dried sprouts on the yolk composition. The paper is well written and provides interesting information. Authors wrote a similar prior article including different information. In general, introduction gives enough information for readers but material and methods, results and discussion need to be improve. In material and methods section, authors should provide further information of some methodologies and on sample collection. Tables have also some mistakes and discussion must be improved in some aspects.
Dear Reviewer 2,
We are sorry that the materials and methods are not exhaustive, in order to avoid overlapping with the previous papers, we have preferred to reduce these important details; however, we have now added the requested things in order to be more comprehensive and make the reading clearer.
Abstract.
Line 14. as well as vitamins or antioxidants compounds (i.e., vitamin A and E).
(i.e., vitamin A and E) this are vitamins, and should be after vitamins. Vitamin A is also considered antioxidant?
A: Yes, retinol (vitamin A) has also antioxidants property, such as their precursor pro-vitamin A (i.e. carotenes). Anyhow, to respect the number of characters we deleted the article (i.e., vitamin A and E)
Line 20. A decrease in oxidative products was observed.
Could you please indicate in which groups? Could you clarify this sentence?
A: Done
Line 32. and other minerals, as 31 well as vitamins or antioxidants compounds (i.e., vitamin A and E)
Move (i.e., vitamin A and E) after vitamins as indicated before or delete it.
A: Done
Materials and Methods
Lines 116-117. For each dietary treatment, 10 pools of 10 egg yolks were collected during weeks 0, 116 4, and 8 and stored at 5 °C until analytical processing (maximum 2 days later).
Please indicate how many eggs per treatment. How many eggs did you collected per day and how many days?. Where eggs collected different days distributed for different determinations? Or did you pool the eggs of different days and make same determinations in the pooled mixture. You should clarify the sample collection information.
A: We added the information required. We collected 9 egg/group/day for a total of 56 days (experimental period) + 10 days of baseline collection (before dietary administration). The egg collection per days required a performance production data management, but as explained in the general comment, considering the low number of animals used, we prefer to report the total production of eggs and not the egg distribution per day, since the number of hens cannot be considered sufficiently reliable for studying the performance and is beyond the scope of the paper. Finally, 100 eggs from the three main periods (baseline, 4, and 8 weeks) have been collected and pooled (10 pools of 10 eggs) for analytical determination. We decided to explain what here reported in the legend of Figure 1.
.Line 149. FD. Did you mean fluorescent detector?.
- yes, we extensively wrote it
Lines 142-151-Did you use a quaternary pump for HPLC of antioxidants?. Please indicate the total running time. Also indicate the solvents used and program.
A: Yes, we use a quaternary pump for HPLC. We detailed the method.
Lines 152-167. Indicate run time for carotenoids. How do you make the sample extraction?. Did you use the same procedure based on saponification described for vitamins?
A: The run time is reported for every programmed step. The extraction is the same of tocopherols. We reported all in the M&M
Linjes 171-190. Indicate how did you make the extraction of phytoestrogens. Did you use the same procedure based on saponification described for vitamins?
A: Done, we reported the method
Lines 192-194. The sterols in the diets and yolks were extracted with n-hexane as reported by Mattioli et al. [16]. Starting from 0.1 g of freeze-dried and finely ground sprouts or yolk.
How did you ground the sample. Particle size?, or apparatous used (vortex?, manually? Etc)
A: Done
Lines 192-204- Please describe briefly the extraction procedure or was the same as described for tocopherols?. Please indicate the run time for sterols, the solvents used for the mobile phase and the flow rate. Also, indicate volumen inyected.
A: The extraction was the same of Tocopherols. The other notes were added.
Lines 213-214. Change D3 by D3 (subíndice). Check though out the text.
A: Done
Lines 228-229. How did you make the extraction from samples. Did you homogenized them?.
A: We use the same procedure of sample preparation for all analytical determination.
RESULTS
Table 1. Maize. Do you mean corn?
A: yes, it is corn
For Feed, Alfalfa and Flax. Indicate abbreviation and define in table foot.
A: Done
Indicate the form of Zn, Fe, I and choline added in the premix.
A: The mineral premix was a commercial multi-vitamins mineral premix, we reported what indicated in the label. For more specification, we added the name of manufacturer (ZAGRO company).
Check subíndices for B1, B2, B6, B12
A: Done
Table 2.
a-tocopheryl acetate instead a-tocopherol acetate
A: The synthetic form present in the mineral premix is alpha-tocopheryl acetate and not a-tocopherol acetate.
Vitamin D in diets is missing
A: There is 3000 IU of cholecalciferol in the mineral mix as specified in the table.
Lines 270-272. Indicate in which table are these data presented
…was higher in the A and F diets than in control. Did you test it statistically. P value is not presented in table
A: Done. Yes, we statistically test all data, but considered that for feed we had just three replicates of the same samples, we retained more correct to report the mean ± SEM, without comparison and P value.
Line 290. Vitamin D3. Check subíndice
A: Done
Table 3 instead Tabella 3.
A: Done
The sum of carotenes it seems include Retinol. May be you can present retinol after the sum of carotenes
A: Lutein, zeaxanthin, and beta-carotene are considered plant precursors of vitamin A, for this reason we classified retinol in carotenes groups (see table), but we did not include it in the carotenes sum.
Check subindices for table 3. Colecalcipherol??. Should be cholecalcipherol?. The same for hydroxycolecalcipherol
A: Done
Table 4 instead Tabella 4.
A: Done
Change Yolk oxidative status of colesterol by “colesterol oxidative products” which is the definition of COPs
A: Done
Change “significance” by P value
A: Done
Define in the foot of the table RMSE and add superindices of definitions provided in foot table.
A: Done
Line 335. D3 correct subíndice
A: Done
Table 5 instead Tabella 5. Cholestol??. Do you mean cholesterol??
A: Done
Figure 1 instead Figure 2. Why did you present this regression between cholesterol and vitamin D?. Cholesterol was negatively correlated with other parameters with lower P value. I would recommend to present a table with the parameters of the regression. Did you check if slopes were statistically different between treatments? You could make this comparison and include information in table that you built including R2 and P of the equation. Because for these parameters correlations were 0.051 but may be regression adjustments was not statistically significant.
A: The reviewer note is correct. We decided to eliminate the figure from the paper.
DISCUSSION
It is missing if all the compounds tested are within values found in literature. Check with references.
Lines 369-371. The Pearson matrix demonstrated that cholesterol was negatively correlated with antioxidants (P = 0.002), phytosterols (P = 0.001), phytoestrogenes (P = 0.016), vitamin D (P = 370 0.05), and n-3 PUFAs (P > 0.05).
But correlation between colesterol and n-3 fatty acids was not statistically significant so there was not correlation between these variables.
A: We corrected the sentence
Lines 397-400. In addition, PUFAs can contribute to the reduction of egg cholesterol [28], although the correlation was non-significant. Our results showed that both sprouts decreased ALA, and increased EPA respect to the Control diet; an increase in DHA was found only in eggs of F group.
Why dont you present long chain PUFA in the composition of the diets?.You give importance to these fatty acids and these were affected by dietary treatments but you only present C18:2 and C18:3.
A: In the feed ingredients, as plant derivative, the LC-PUFA are not present. More than 18 carbon atoms are only present in algae, as marine products they are the only vegetable source of LC-PUFA.
Lines 407-409. This study confirms the efficiency of hens to elongate and desaturate ALA into n-3 407 LC-PUFAs [34,35] and to transfer it into the eggs, where they play a crucial role during 408 chick embryo development [36].
You should include the desaturase index to affirm this. Why are you sure these were not from the diets?.
A: Because in the diets the LC-PUFA are not present (see the previous point), what is found in the eggs is exclusively due to the direct elongation and desaturation activity of the hens. Furthermore, we retain that the desaturase index cannot be applied in nutritional studies, because, being an estimation of fatty acids precursor/derivatives (and not a direct enzyme quantification), it is obviously modulated by dietary fatty acids and it did not represent a “real” metabolic image of the animals.
Could be these higher long PUFA proportion found by sprouts due to higher antioxidant contents and protection. Clarify.
A: Yes, the reviewer’s note is correct. The higher LC PUFA presence could be due to the preservation activity of sprouts antioxidants, we added this aspect in discussion.
Lines 432-442. Tocols and carotenes are relevant antioxidant molecules often combined to exploit 432 their different properties. α-Tocopherol, the most abundant vitamin E isoform, has a 433 chain-breaking action and can be regenerated from its oxidized form by ascorbic acid, 434 carotenoids, and phospholipids……………..
Please, include which treatment do you think is better to improve antioxidants content.
A: Done
How do you explain the time x diet interaction?. Retinol and tocols decrease in treatment A with time but increased in F. Could be this effect by interaction between different component??. Review in literature.
A: Done
Lines 443-446. Hens fed freeze-dried sprouts exhibited reduced lipid oxidation and COPs. Indeed, 443 the amount of antioxidants was negatively correlated with COPs, whereas the correlation 444 between cholesterol and COPs was positive, probably because the COP decrease was also 445 due to the reduction of cholesterol and thus attributed to substrate limitation [42].
Include more information of the negative effects of COPS
A: Done
Lines 447-449. The increase in total vitamin D in eggs of hens fed freeze-dried sprouts was also no- 447 table. From a research perspective, eggs have garnered significant attention as vehicles for vitamin D bio-fortification or enrichment [4]. Eggs naturally contain vitamin D and are 449 thought to contribute up to 10% of dietary intake [9]. Raw eggs contain approximately 3 450 μg of vitamin D per 100 g of egg [9,43], w……
Include recommended values of vitamin D for humans.
A: Done
Lines 455-456. Linear regression of vitamin D and cholesterol demonstrated that sprout treatment 455 reduced cholesterol content and increased vitamin D, especially when flaxseed was ad- 456 ministered
How do you explain this effect?
A: Body of literature, reported that the cholesterol is a precursor of Vitamin D, thus, it is possible that the reduction of the cholesterol in egg yolk was due to metabolic processes occurred in the hens liver, with the conversion in the latter (Barnkob, L.L.; Argyraki, A.; Jakobsen, J. Naturally enhanced eggs as a source of vitamin D: A review. Trends Food Sci. Technol. 2020, 102, 62–70.). We exposed this concept in the discussion.
Reviewer 3 Report
This study is designed to investigate the effects of supplementing freeze-dried flaxseed and alfalfa sprouts on the contents of the ingredients in chicken eggs, especially the contents of bioactive compounds. The manuscript is well written, however, the following concerns should be addressed before the manuscript is considered for publication.
1. Eggs are a complete food with many nutritional ingredients, including protein, fat, micro- or trace-elements, bioactive compounds, etc. Why did the authors just determine the contents of a few kinds of nutrients? I think it is also important to know how the contents of other nutrients are affected by dietary freeze-dried flaxseed and alfalfa sprouts. Please provide an explanation.
2. Why did the authors just determine the contents of the main polyunsaturated fatty acids in an egg? How about the saturated and mono-unsaturated fatty acids, such as C16:0, C18:0, and C18:1n-9 fatty acids? Please provide an explanation.
3. The also should provide the content of total fat in the eggs.
4. I suggest that the authors provide the information on the effects of dietary freeze-dried flaxseed and alfalfa sprouts on the production performance of the laying hens, such as the egg production, the egg laying rate.
5. The table 3 and 4 should use the three-line table format.
6. In the caption of Table 3, ‘main fatty acids’ should be ‘main polyunsaturated fatty acids’.
7. A minor editing work on text is needed. For example, ‘Tabella 3’ should be ‘Table 3’.
Round 2
Reviewer 2 Report
The paper has now improved considerably. Only minor mistakes were found as follows:
Table 1. Please include C, A, and F in parenthesis in the heading of table 1
Table 1. The form of Zn (zinc oxide??), Fe (ferrous sulphate??), I. Should be included in the premix composition
Table 1. Check subindex for B12
Table 2- a-tocopheryl acetate
Line . Decreased instead decrease
Reviewer 3 Report
Comments and Suggestions for Authors
The authors responded to my comments, and almost addressed all my concerns. The following is my new comments on the authors’ responses:
1. A: We also evaluated the macronutrient profile; however, no differences were recorded, therefore, considering the scientific profile of the journal (Antioxidant Molecules) and the numerous molecules analyzed, we decided to focus the paper on antioxidants, fatty acids, phytoestrogens, vitamin D, and phytosterols. We have, however, added the yolk lipid profile to Table 4 as suggested later.
Comment: in the manuscript, the authors should mention they evaluated the macronutrient profile of the eggs and no significant differences were found in which macronutrients. If possible, the authors may present it as supplementary materials.
2. A. Obviously, we looked at the complete fatty acid profile, but we reported only those that are significantly different for the same reason explained before. There are already too many tables in the manuscript, and we had to reduce them, merging some traits for rendering the document clear and readable.
Comment: in the manuscript, the authors should mention they evaluated the complete fatty acid profile of the eggs and no significant differences were found in which fatty acids. If possible, the authors may present it as supplementary materials.
3. A: We added the lipid % in table 4.
Comment: good.
4. A. Unfortunately, we cannot use the production performance data in the manuscript, because the experimental design was designed for a different scope (see general comment). However, we have included one table for the reviewer's information.
Comment: good. If possible, the authors may present it as supplementary materials.
5. A: Done
Comment: good.
6. A: Done
Comment: good.
7. A: We carefully checked the paper for minor mistakes.
Comment: good.
